# Preparation and Characterization of Nano-Silver-Loaded Antibacterial Membrane via Coaxial Electrospinning

**DOI:** 10.3390/biomimetics8050419

**Published:** 2023-09-11

**Authors:** Qingxi Hu, Zhenwei Huang, Haiguang Zhang, Murugan Ramalingam

**Affiliations:** 1Rapid Manufacturing Engineering Center, School of Mechatronic Engineering and Automation, Shanghai University, Shanghai 200444, China; huqingxi@shu.edu.cn (Q.H.); hzzzzzw@163.com (Z.H.); 2Shanghai Key Laboratory of Intelligent Manufacturing and Robotics, Shanghai University, Shanghai 200072, China; 3National Demonstration Center for Experimental Engineering Training Education, Shanghai University, Shanghai 200444, China; 4IKERBASQUE, Basque Foundation for Science, 48013 Bilbao, Spain; rmurug2000@gmail.com; 5Joint Research Laboratory (JRL), Faculty of Pharmacy, University of the Basque Country (UPV/EHU), 01006 Vitoria-Gasteiz, Spain

**Keywords:** coaxial electrospinning, process parameters, membrane, nanofibers, antibacterial, biomedical, vascular

## Abstract

The coaxial electrospinning process has been widely used in the biomedical field, and its process parameters affect product quality seriously. In this paper, the influence of key process parameters of coaxial electrostatic spinning (solution concentration, electrospinning voltage, acceptance distance and liquid supply velocity) on the preparation of a membrane with Chitosan, Polyethylene oxide and nano-silver as the core layer and Polycaprolactone as the shell layer was studied. The optimal combination of key process parameters was obtained by using an orthogonal test, scanning electron microscope, transmission electron microscope and macro-characterization diagram. The results showed that the coaxial electrospun membrane had good mechanical properties (tensile strength is about 2.945 Mpa), hydrophilicity (the water contact angle is about 72.28°) and non-cytotoxicity, which was conducive to cell adhesion and proliferation. The coaxial electrospun membrane with nano-silver has an obvious inhibitory effect on Escherichia coli and Staphylococcus aureus. In summary, the coaxial electrospun membrane that we produced is expected to be used in clinical medicine, such as vascular stent membranes and bionic blood vessels.

## 1. Introduction

Coaxial electrospinning is a unique method of producing nanofiber [1]. The nanofiber membranes prepared by coaxial electrospinning have more surface area, higher porosity and the capacity to simulate the structure and biological function of the natural extracellular matrix [2]. Coaxial electrospun nanofibers provide the advantage of focused drug delivery, gradual drug release, high transfection efficiency, quick onset of action and favorable pharmacokinetics when used in drug delivery systems [3]. Researchers can design nanofibers with any desired properties, such as improved biocompatibility, drug loading performance or mechanical properties, through coaxial electrospinning. Therefore, coaxial electrospinning has caught the interest of numerous researchers in the field of biomedicine and has found successful applications in drug loading, wound repair, the development of anti-inflammatory and antibacterial agents, biological tissue engineering and other areas.

Numerous variables, including those related to the solution’s viscosity, concentration and interactions with other solutions, as well as process variables like electrospinning voltage, receiving distance and core–shell solution liquid supply velocity and environmental variables like humidity and temperature, all play a role in the formation of core–shell nanofiber structures during coaxial electrospinning [4,5,6]. Therefore, it is typically necessary to carefully optimize each process parameter and then carry out orthogonal test analysis on them after obtaining the appropriate value of each process parameter in order to obtain the optimal combination of process parameters so as to ensure the smooth progress of coaxial electrospinning.

The membrane is expected to be used in medical clinics in the future, so the membrane should be made of biological materials with good biocompatibility. Polycaprolactone (PCL), an organic polymer, is a synthetic biomaterial approved by the Food and Drug Administration (FDA) for use in drug loading and tissue engineering [7]. PCL has good mechanical properties and biocompatibility, but it has strong hydrophobicity, which is not conducive to cell adhesion. Therefore, in order to obtain better cell adhesion properties, it is necessary to add some hydrophilic materials to use with PCL to improve the hydrophilicity of the membrane [8,9,10]. Chitosan (CS) is a modified natural polymer derived from chitin, which has high biocompatibility and hydrophilicity, and the addition of CS can improve the hydrophilicity of the membrane. However, its spinnability is limited due to the polycationic properties in solution, rigid chemical structure and special molecular interactions [11,12,13]. Polyethylene oxide (PEO) is a crystalline, thermoplastic water-soluble polymer with good hydrophilicity and biocompatibility. It has been reported that it has no antigenicity and only low toxicity, but it can improve the spinnability of CS, so it can be added as a co-spinning agent to the preparation of the membrane [14,15]. Nano-silver is a metallic silver substance with a particle size of nano-scale, which has a broad-spectrum antibacterial effect and is an excellent antibacterial drug [16,17]. Due to its low toxicity and good antibacterial activity, it is often used in medicine for blood vessel coating and biological patches [18]. When preparing the membrane, the addition of nano-silver can make the membrane have antibacterial properties. Moreover, when nano-silver is used together with CS, CS can also act as a stabilizer to prevent the aggregation of nano-silver [19].

Nowadays, vascular stents and bionic blood vessels are often used in clinical interventional therapy to treat various vascular diseases [20]. However, due to their lack of antibacterial function, infections often occur during the operation or even several months after the operation, which seriously affects the recovery of patients. Actually, some even need a second open operation for infection treatment, which is undoubtedly a burden for patients [21]. Therefore, it has become an urgent problem in clinical medicine to prepare a kind of membrane that can be combined with the existing vascular stent and bionic blood vessels to make them have antibacterial function.

The aim of this paper is to study the preparation of a membrane with antibacterial function by using coaxial electrospinning with PCL as a shell layer and CS, PEO and nano-silver as the core layer. The ideal combination of essential process factors was achieved by using an orthogonal test after extensive investigation of numerous process parameters, and the physicochemical features of the produced membrane were described. The biocompatibility and antibacterial characteristics of the coaxial electrospun membrane are investigated. The coaxial electrospun membrane promotes cell attachment and proliferation while also inhibiting the growth of *Escherichia coli* (*E. coli*) and *Staphylococcus aureus* (*S. aureus*). This coaxial electrospun membrane may be used as a coating of the vascular stent and the bionic blood vessel for antibacterial function in the future.

## 2. Materials and Methods

### 2.1. Materials for Preparation of Coaxial Electrospun Membrane

PCL (Mn = 80,000 g/mol) was purchased from Miracll Chemicals Co., Ltd. (Shanghai, China); the solution of Silver nanoparticles (concentration: 10 g/L) was purchased from Mingcheng Plastic Additives Chemical Co., Ltd. (Dongguan, China). Sinopharm Chemical Reagent Co., Ltd. (Shanghai, China) supplied the following chemicals: CS (molecular weight: roughly 300 kDa), PEO (molecular weight: approximately 600 kDa), Dichloromethane (DCM), N,N-Dimethylformamide (DMF) and acetic acid; human umbilical vein endothelial cells (HUVECs) was provided by Changhai Hospital. No materials had undergone any alteration or further purification.

### 2.2. Analysis of Key Process Parameters of Coaxial Electrospinning

Preparation of electrospinning solution: Firstly, we prepared the shell electrospinning solution. As the shell layer of the nanofibers only needs to provide good mechanical properties for the coaxial electrospun membrane, PCL was chosen; at room temperature, the mixed solution of DMF and DCM (DMF:DCM, 7:3) was prepared, the PCL particles were weighed and dissolved in the mixed solution, the beaker mouth was sealed, and the 15% (*w*/*v*) PCL solution was obtained as the shell solution after stirring for 4 h. Secondly, we prepared the core electrospinning solution. As the core layer of the nanofibers, it needs to provide better hydrophilicity, biocompatibility and drug-loading properties; it is difficult to obtain these properties from a single material, so a mixture of multiple materials was chosen (CS and PEO). The CS powder with different weights was weighed and dissolved in 70% acetic acid, and 2% (*w*/*v*), 3% (*w*/*v*), 4% (*w*/*v*) and 5% (*w*/*v*) CS solutions were obtained; the PEO powder with different weights was weighed and dissolved in 70% acetic acid, and 2% (*w*/*v*) CS and 3% (*w*/*v*), 4% (*w*/*v*) and 5% (*w*/*v*) PEO solutions were obtained. After stirring for 6 h, we mixed the same concentration of the CS solution and PEO solution to obtain the mixed solution (CS-PEO solution); the nano-silver solution was added to the CS-PEO solution, which was used as the core solution after stirring for 4 h, and the solution preparation process is shown in Figure 1.

Solution concentration: in the case of other electrospinning process parameters being unchanged, the shell solution is preliminarily determined as a PCL solution with a concentration of 15% (*w*/*v*), with a core solution of 2% (*w*/*v*) CS + 2% (*w*/*v*) PEO, 3% (*w*/*v*) CS + 3% (*w*/*v*) PEO, 4% (*w*/*v*) CS + 4% (*w*/*v*) PEO and 5% (*w*/*v*) CS + 5% (*w*/*v*) PEO.

Electrospinning Voltage: in the case of other electrospinning process parameters being unchanged, the electrospinning voltage is set to 10 kV, 14 kV, 18 kV and 22 kV pre-experiment.

Receiving distance: in the case of other electrospinning process parameters being unchanged, the receiving distances of 12 cm, 15 cm, 18 cm and 21 cm are initially selected pre-experiment.

Liquid supply velocity: In the case of other electrospinning process parameters being unchanged, with a drum receiver (length: 200 mm; diameter: 76 mm), the speed is set to 600 rpm, the shell liquid supply velocity is set as 0.6 mL/h, and the ratio of the nuclear layer to shell solution liquid supply velocity is set as 1:1, 1:2, 1:3 and 1:4 for pre-experimental analysis. The preparation process is shown in Figure 1.

### 2.3. Orthogonal Experimental Analysis

The key process parameters of coaxial electrospinning were optimized through orthogonal experiments. The four experimental factors are (A) solution concentration, (B) electrospinning voltage, (C) receiving distance and (D) liquid supply speed; three levels of each factor were set for the orthogonal test, and the tensile strength was used as the test index. The orthogonal experiment was carried out with the four factors and three levels.

Taking experimental factor A as an example gives the calculation of K_Bi_, K_Ci_, K_Di_, k_Bi_, k_Ci_ and k_Di_ of factors B, C, D and others.

The influence of level 1 of factor A on each experimental group can be obtained as follows:K_A1_ = y_1_ + y_2_ + y_3_, k_A1_ = K_A1_/3

The following formula can be obtained for the influence of level 2 of factor A on each experimental group:K_A2_ = y_4_ + y_5_ + y_6_, k_A2_ = K_A2_/3

The following formula can be obtained for the influence of level 3 of factor A on each experimental group:K_A3_ = y_7_ + y_8_ + y_9_, k_A3_ = K_A3_/3
where K_Ai_ is the sum of experiment indexes, k_Ai_ is the average value of the sum of experiment indexes, and y_i_ is the number of experimental groups.

The degree of influence of each factor on the test results is judged by the size of range R:R = k_max_ − k_min_
where R is the variation amplitude of the test index within the value range of this factor.

### 2.4. Microscopic Characterization of Coaxial Electrospun Membrane

Scanning electron microscope (SEM) analysis: The prepared coaxial electrospun membranes were cut into 10 mm× 10 mm samples and fixed on the copper grids, and then the samples were sprayed with gold with a thickness of about 10–15 nm. Finally, samples were placed on the work platform of SEM (Hitachi SU-1500, HITACHI, Tokyo, Japan) for observation.

Transmission electron microscope (TEM) analysis: In the process of coaxial electrospinning, the 230-mesh carbon-containing copper grid was clamped by using small tweezers with insulated gloves on both hands, and the nanofibers were collected by slightly shaking for 1 min at a distance of 15 cm from the coaxial needle. Then, the copper grid with collected nanofibers was placed on the work platform of TEM (Hitachi HT-7800, HITACHI, Tokyo, Japan), and TEM images were taken to observe the core–shell structure of nanofibers.

### 2.5. Test of Tensile Strength

Rectangular samples of 150 mm × 10 mm were cut out of the membranes prepared with various coaxial electrostatic spinning process parameters, and each sample was gripped with jigs at both ends of the material testing machine with a distance of 50 mm between the two ends. Moreover, a preloading force of 0.1 kN was applied, as shown in Figure 2. The sample was then stretched at a speed of 10 mm/min until it broke, reading the data of each stretch. The above test was repeated 5 times for each sample, and the average value was taken as the result.

### 2.6. Fourier Transform Infrared Spectrometry (FTIR) Analysis

To analyze the interaction between materials, the chemical composition and functional groups of nano-silver, PEO, CS, PCL and the coaxial electrospun membrane were analyzed through FTIR (Avatar 370, Thermo Fisher Scientific, Waltham, MA, USA). Tests were performed in attenuated total reflection (ATR) mode with a spectral resolution of 2 cm^−1^ and a wavenumber range of 4000 cm^−1^–500 cm^−1^.

### 2.7. Test of Hydrophilicity

The hydrophilicity of membranes was evaluated by measuring their water contact angle [22]. The pure PCL membrane prepared by electrospinning was taken as the control group, and the coaxial electrospun membrane was taken as the experimental group. Samples of 1 cm × 1 cm were cut out from the two groups of the membrane, and the sample was placed on the slide and placed on the work table of the water-contact-angle-measuring instrument (OCA 15EC, Dataphysics Ltd., Filderstadt, Germany). A total of 3 µL of deionized water droplets was placed on the membrane, and the images were taken with a CCD camera.

### 2.8. Test of Cell Cytotoxicity

The Cell Counting Kit-8 (CCK-8) is often used to evaluate the toxicity of samples due to its high sensitivity and ease of operation [23]. First, the coaxial electrospun membrane with a nano-silver content of 2, 3 and 4 wt% was placed in the 96-well plate, and the PCL membrane was used as the blank control group. A 100 µL cell suspension of HUVECs with a density of 1 × 10^5^ cells/mL was then inoculated into the 96-well plate. When incubated in the incubator for 1, 3 and 5 days, 10 µL of CCK-8 was added to each well and cultured in the incubator for 3 h. The OD values were obtained at a wavelength of 450 nm by using a microplate reader (Infinite 200 Pro, Tecan Group Ltd., Männedorf, Switzerland).

### 2.9. Test of Cell Adhesion

In order to test the biocompatibility of the coaxial electrospun membrane with a silver content of 2 wt%, 3 wt% and 4 wt%, the activity of cells was detected by using live/dead staining tests. Samples of inoculated HUVECs (1 × 10^5^ cells/mL) were incubated in a constant temperature incubator for 2 and 4 days. The solution was prepared with calcein, propyl iodide and Pbs, and the volume ratio was 1:1:1000. The live dye solution was dropped onto the sample membrane with a pipette gun and cultured in a constant temperature incubator for 15 min. Finally, the sample membrane was placed under an inverted fluorescence microscope to observe the cell activity.

### 2.10. Test of Bacteriostatic Performance

In this experiment, the inhibitory activity of the coaxial electrospun membrane against *E. coli* (representative of Gram-negative bacteria) and *S. aureus* (representative of gram-positive bacteria) was tested in vitro. First, the coaxial electrospun membrane containing nano-silver was cut into a sample size of 1 cm × 1 cm, and the PCL membrane without nano-silver was also cut into a sample size of 1 cm × 1 cm. Secondly, the coaxial electrospun membranes were placed at the bottom of each culture dish in the super clean table, a PCL membrane was also placed as a control group, and then the surface was evenly coated with 30 µL of *E. coli* or 30 µL of *S. aureus*, sealed with a sealing strip and stored in a 37 °C incubator. The inhibition area was observed at 72 h.

## 3. Results and Discussion

### 3.1. Influence of Key Process Parameters of Coaxial Electrospinning on Membrane

Solution concentration: During coaxial electrospinning, the concentration of the core layer and the shell layer directly affects the forming quality of the nanofibers [24]. As shown in Figure 3, when the solution concentration was 2% (*w*/*v*), the unstable Taylor cone could be formed at the needle, often showing a whipping phenomenon, leading to the formation of more beads, and the forming quality of nanofibers was poor; the diameter distribution of nanofibers was wide, with an average diameter of 538 ± 184 nm. When the solution concentration was 3% (*w*/*v*), the Taylor cone at the needle was stable, the diameter distribution was more uniform, and the forming quality was better; the diameter of the nanofibers was evenly distributed, with an average diameter of 551 ± 122 nm. When the solution concentration was 4% (*w*/*v*), the electric field force had difficulty breaking through the surface tension of the solution and could not form a stable Taylor cone, and the nanofibers showed uneven thickness and adhesion, with an average diameter of 602 ± 261 nm. When the solution concentration was 5% (*w*/*v*), the nozzle showed intermittent blockage, the forming quality was poor, and the electrospinning could not be sustained. The experimental results show that when the solution concentration is about 3% (*w*/*v*), the forming quality of nanofibers is better and the diameter distribution of nanofibers is more uniform.

Electrospinning Voltage: The diameter and forming quality of nanofibers are directly affected by the electrospinning voltage during coaxial electrospinning, and the electrospinning voltage directly affects the stability of the Taylor cone [25,26]. As shown in Figure 4, when the voltage was 10 kV, although there was a jet formation at the needle, because the solution could not completely overcome the surface tension, it could not form a stable Taylor cone and nanofibers showed more beading defects; the diameter distribution of nanofibers was wide, with an average diameter of 538 ± 184 nm. When the voltage was 14 kV, the Taylor cone at the needle was stable, the forming quality was high, and the diameter of nanofibers was evenly distributed, with an average diameter of 560 ± 96 nm. When the voltage was 18 kV, the Taylor cone whipped, the electric field force was too large, the solution at the needle burst, the forming quality decreased, and the diameter distribution of the nanofibers was uneven, with an average diameter of 584 ± 220 nm. When the voltage was 22 kV, the Taylor cone was very unstable, the fiber had a lot of adhesion and accumulation, and the forming quality was poor, with an average diameter of 558 ± 315 nm. The experimental results show that the forming quality of nanofibers and the diameter distribution of nanofibers are better when the electrospinning voltage is about 14 kV.

Receiving distance: The receiving distance refers to the distance between the electrospinning needle and the receiver, which affects the degree of solvent volatilization during the stretching and flight of the solution [26]. As shown in Figure 5, when the receiving distance was 12 cm, the solvent volatilization in the solution was incomplete, resulting in a large number of bead defects, and the diameter distribution of nanofibers was not uniform, with the average diameter of 532 ± 243 nm. When the receiving distance was 15 and 18 cm, the jet was not overstretched, the solvent was well volatilized, the nanofibers were evenly distributed, and the forming quality was good, with an average diameter of 572 ± 107 nm and 550 ± 106 nm. When the receiving distance was 21 cm, the jet was overstretched, resulting in more beading defects in the fibers and unsatisfactory collection of the fibers, and the diameter distribution of nanofibers was not uniform, with an average diameter of 663 ± 259 nm. The experimental results show that the forming quality of nanofibers and the diameter distribution of nanofibers are better when the receiving distance is about 15–18 cm.

Liquid supply velocity: In coaxial electrospinning, the liquid supply velocity of the shell solution and core solution directly affects the forming effect of nanofibers with a core–shell structure. As shown in Figure 6, when the ratio of core–shell liquid supply velocity was 1:1, the liquid supply velocity of the core layer was obviously too fast, the core layer solution broke through the viscous stress of the shell solution, and the nanofibers with a core–shell structure could not be prepared. When the ratio of core–shell liquid supply velocity was 1:2, the nanofibers with core–shell structure could be prepared, but the core–shell boundary was not obvious. When the ratio of core–shell liquid supply velocity was 1:3, the core–shell liquid supply velocity was appropriate, the Taylor cone was stable, and the core–shell boundary was obvious. When the ratio of core–shell liquid supply velocity was 1:4, the liquid supply velocity of the core layer was slow. Although the nanofibers with a core–shell structure could be prepared, the Taylor cone was unstable in the electrospinning process, and the spinning disorder of the shell solution occurred intermittently.

### 3.2. Analysis of Orthogonal Test Results

As can be seen from Table 1, among the four key process parameters, solution concentration had the greatest influence, followed by electrospinning voltage and the ratio of core–shell liquid supply velocity, and receiving distance had the least influence. The optimal combination of key process parameters obtained by the experiment was A_2_B_1_C_3_D_3_, A_2_B_2_C_2_D_2_ and A_2_B_3_C_1_D_1_. Combined with their macro-characterization diagram, it can be seen from Figure 7 that A_2_B_2_C_2_D_2_ had the best membrane surface quality; that is, the solution concentration was 3% (*w*/*v*) CS + 3% (*w*/*v*) PEO, the electrospinning voltage was 14 kV, the receiving distance was 16 cm and the core–shell liquid supply velocity ratio was 1:3, which was the best combination of process parameters, and the coaxial electrospun membrane prepared with this combination would achieve the best mechanical properties (tensile strength: 2.945 ± 0.092 MPa).

### 3.3. Fourier Transform Infrared Spectrometry (FTIR)

Figure 8 shows the FTIR spectra of each material and the coaxial electrospun membrane. The infrared spectrum of pure PEO shows a peak at 1098 cm^−1^, which is attributed to the stretching vibration of the ether group (C-O-C) [27]; there is a strong and broad peak at 3459 cm^−1^ in the infrared spectrum of pure CS, which is related to O-H and N-H stretching bond [13]. In the PCL infrared spectrum, the peak at 2947 cm^−1^ is for the C-H stretch, and the strong peak at 1727 cm^−1^ is for the C=O double bond [28]. Thus, the infrared spectrum of the coaxial electrospun membrane shows a characteristic peak at 1099 cm^−1^, which is related to the C-O-C stretching vibration of PEO, and a peak at 1729 cm^−1^, which is related to the C=O carbonyl stretching of PCL; the presence of CS in the coaxial electrospun membrane is confirmed by the broad bond centered around 3450 cm^−1^ due to O-H and N-H stretching and the presence of a characteristic peak at 1468 cm^−1^ due to the CH_2_ bending of polysaccharide. The presence of a characteristic peak at 2351 cm^−1^ confirmed that nano-silver is physically adsorbed [29].

In addition, the main characteristic peaks of DCM are at 3000 cm^−1^–2900 cm^−1^ and 1400 cm^−1^–1300 cm^−1^, corresponding to the stretching vibration of C-H and C=O, respectively [30]; the main characteristic peaks of DMF are at 3000 cm^−1^–2800 cm^−1^ and 1700 cm^−1^–1600 cm^−1^, corresponding to the stretching vibration of C-H and C=O, respectively [31]. The above characteristic peaks of DCM and DMF are not found in the infrared spectrum of the coaxial electrospun membrane, and it can be preliminarily inferred that there are no remnants of these two solvents (DCM and DMF).

### 3.4. Hydrophilicity

Hydrophilicity is an important index used to evaluate the biocompatibility of the membrane, and it can be judged from the water contact angle [32]. Good hydrophilic film is conducive to cell adhesion and proliferation, but a too-good hydrophilic film will also affect cell migration [33]. As can be seen from Figure 9, the water contact angle of the pure PCL membrane was 125.22°, showing strong hydrophobicity. Due to the good hydrophilicity of CS and PEO, after adding CS and PEO for coaxial electrospinning, the hydrophilicity of the coaxial electrospun membrane was significantly improved, and the water contact angle of the coaxial electrospun membrane was 72.28°. Compared with electrospinning with PCL alone [34], the addition of CS and PEO could effectively improve the hydrophilicity of the membrane.

### 3.5. Cytotoxicity

Nano-silver has antibacterial activity, but in high concentrations, it causes toxicity to human cells and even harms human health [35]. As shown in Figure 10, after 1, 3 and 5 days of cultivation, OD values of the four groups increased steadily. At 1 d, there was no statistically significant difference between the four groups. At 3 d, the cell proliferation rate in the PCL membrane group and HUVECs group was faster, and the survival rate of coated cells with a nano-silver content of 4 wt% was lower than that of other groups. The cell survival rate of the PCL membrane group and HUVECs group was still higher than that of nano-silver-containing membrane groups until 5 d, the cell survival rate of the membrane with a nano-silver content of 4 wt% was still lower than that of other groups, and the cell survival rates of the coaxial electrospun membranes with a nano-silver content of 2 wt% and 3 wt% were similar, only slightly lower than HUVECs group. The results showed that the coaxial electrospun membrane with nano-silver had almost no obvious inhibitory effect on HUVECs and only slight cytotoxicity when the content of nano-silver was increased to 4 wt%.

### 3.6. Cell Adhesion

As shown in Figure 11, the distribution of living and dead cells after culture for 2 and 4 days was observed, and the quantitative statistics of cell viability are shown in Figure 12. After 2 days, the number of adherent cells on the coaxial electrospun membrane was similar in all groups, there were almost no dead cells in the group with a nano-silver content of 2 wt%, and the viability of cells was high with a value of 96.01 ± 0.14%, while there were a few dead cells in the groups with a nano-silver content of 3 wt% and 4 wt%, with a cell viability of 88.68 ± 0.79% and 83.68 ± 1.17%, respectively. On the 4th day, the number of cells on both membranes showed a steady growth trend, and a small number of dead cells appeared in the group with a nano-silver content of 2 wt% and 3 wt%. The cell viability of these two groups was high, with values of 96.23 ± 0.10% and 95.91 ± 0.11%, respectively; however, more dead cells appeared in the group with a nano-silver content of 4 wt%, and the cell viability here was 89.61 ± 0.59%, indicating that the high nano-silver content caused toxicity to cells. After 4 days, the adhesion and proliferation of cells on the membrane showed a steady growth trend. In the group with a nano-silver content of 2 wt% and 3 wt%, fewer dead cells appeared, which may be due to the anoxic death of the underlying cells caused by excessive cell density rather than the toxicity caused by the addition of nano-silver to the coaxial electrospun membrane.

### 3.7. Bacteriostatic Performance

In this experiment, the inhibitory activity of the coaxial electrospun membrane against *E. coli* and *S. aureus* was tested in vitro, which were representative of gram-negative and gram-positive bacteria respectively [36,37,38]. Due to the addition of nano-silver, the coaxial electrospun membranes has antibacterial effect. As can be seen from Figure 13, the sample on the left side of the culture dish is coaxial electrospun membrane, and the sample on the right side is control group, when the concentration of nano-silver was 3 wt%, the inhibition area in both *E. coli* and *S. aureus* was obvious, and the inhibition area still existed within 72 h, while the inhibition area did not appear in the membrane without nano-silver, indicating that it had no effect on bacterial reproduction. It was also preliminarily proved that the coaxial electrospun membrane containing nano-silver had obvious inhibitory effect on the representative of gram-negative bacteria (*E. coli*) and gram-positive bacteria (*S. aureus*). Due to the bactericidal mechanism of nano-silver, after killing bacteria, nano-silver will be freed from bacteria and continue to produce antibacterial effect [39], so it can be preliminarily inferred that nano-silver released by the coaxial electrospun membrane in the lesion area has a long-term antibacterial effect.

## 4. Conclusions

In this study, CS, PEO and nano-silver were used as the core layer, and PCL was used as the shell layer. The antibacterial membrane of nanofibers with coaxial structure was successfully prepared by coaxial electrospinning process. The analysis of key process parameters and orthogonal test showed that the optimal combination of key process parameters of coaxial electrospinning was as follows: The solution concentration is 3% (*w*/*v*) CS + 3% (*w*/*v*) PEO + 3 wt% nano-silver, the electrospinning voltage is 14 kV, the acceptance distance is 16 cm, and the core–shell liquid supply velocity is 1:3. According to water contact angle test, the coaxial electrospun membrane has good hydrophilicity; The results of cell test in vitro showed that the coaxial electrospun membrane was beneficial to the adhesion and proliferation of HUVECs; The bacteriostatic test showed that the coaxial electrospun membrane have obvious inhibition effect on *E. coli* and *S. aureus*. The coaxial electrospun membrane was expected to be used as an antibacterial coating for vascular stents and bionic blood vessels.

## Figures and Tables

**Figure 1 biomimetics-08-00419-f001:**
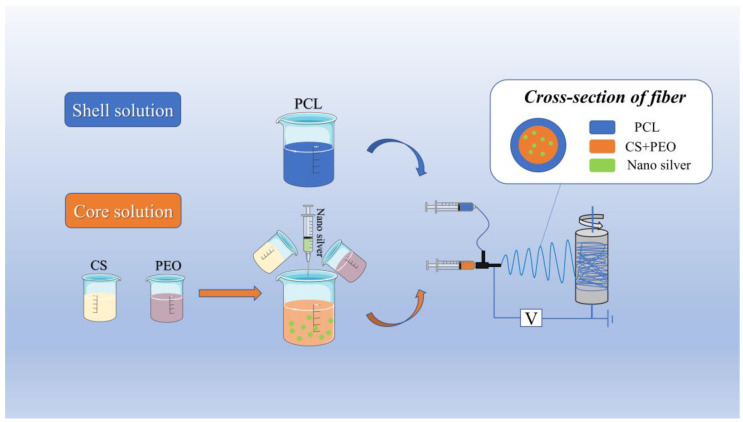
Flow chart of preparation of coaxial electrospun membrane.

**Figure 2 biomimetics-08-00419-f002:**
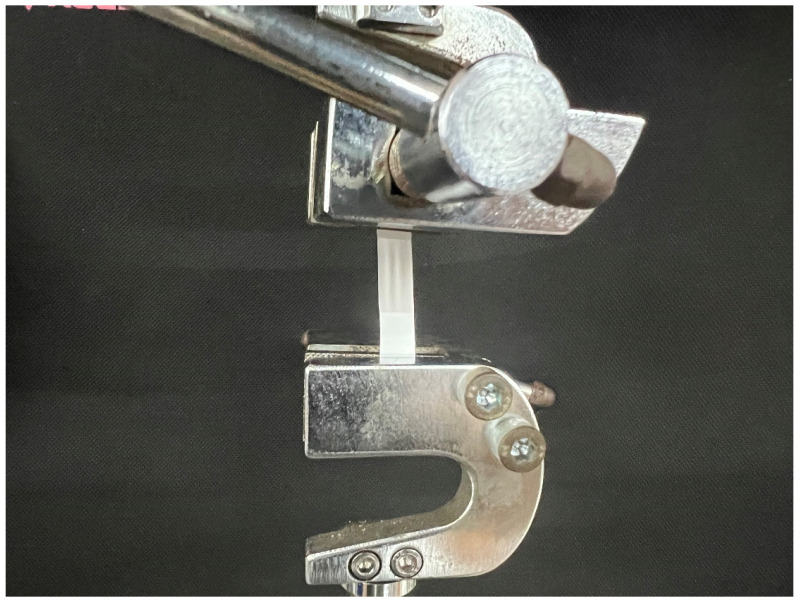
Schematic diagram of installation for tensile test of tensile strength.

**Figure 3 biomimetics-08-00419-f003:**
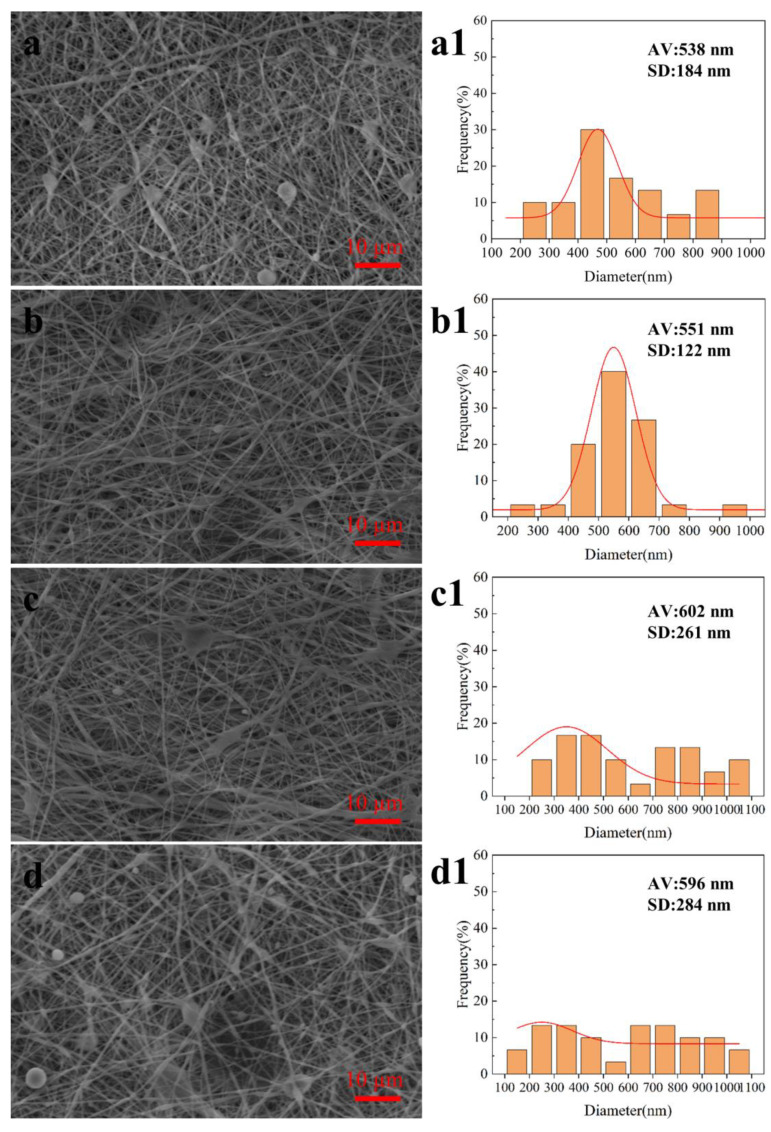
SEM images and fiber diameter distribution of coaxial electrospun membranes prepared with different solution concentrations. (**a**,**a1**) 2% (*w*/*v*); (**b**,**b1**) 3% (*w*/*v*); (**c**,**c1**) 4% (*w*/*v*); (**d**,**d1**) 5% (*w*/*v*).

**Figure 4 biomimetics-08-00419-f004:**
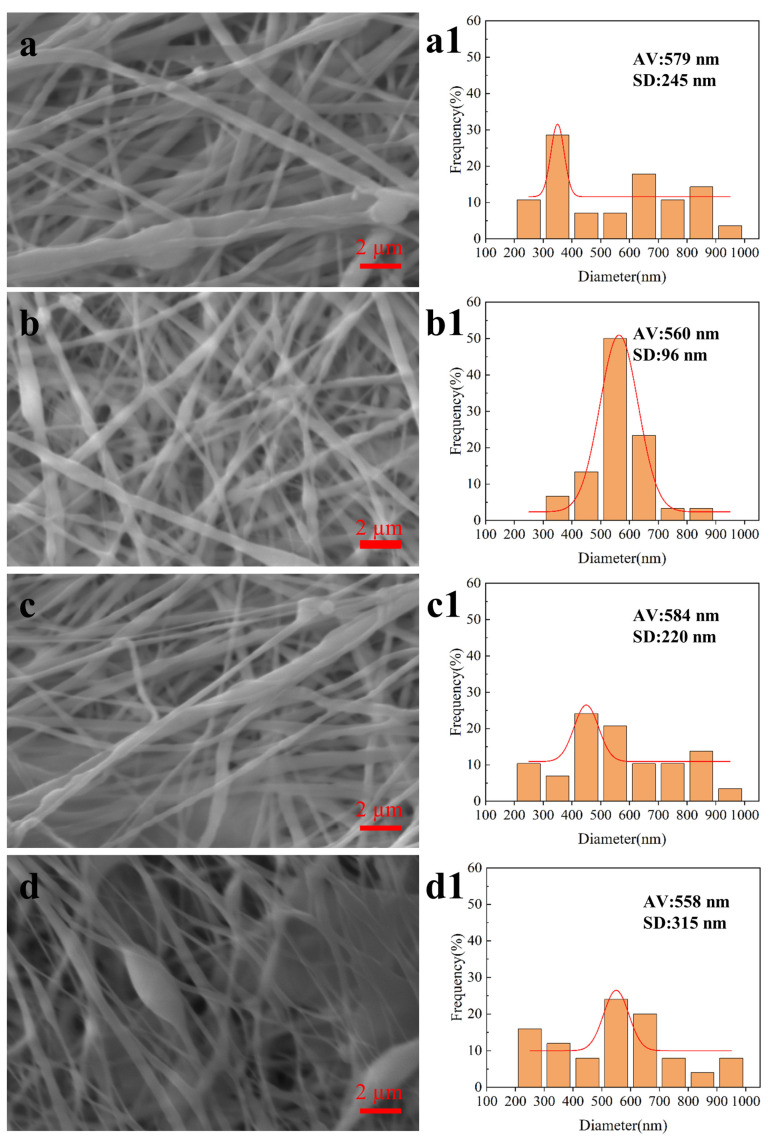
SEM images and fiber diameter distribution of coaxial electrospun membranes prepared with electrospinning voltage. (**a**,**a1**) 10 kV; (**b**,**b1**) 14 kV; (**c**,**c1**) 18 kV; (**d**,**d1**) 22 kV.

**Figure 5 biomimetics-08-00419-f005:**
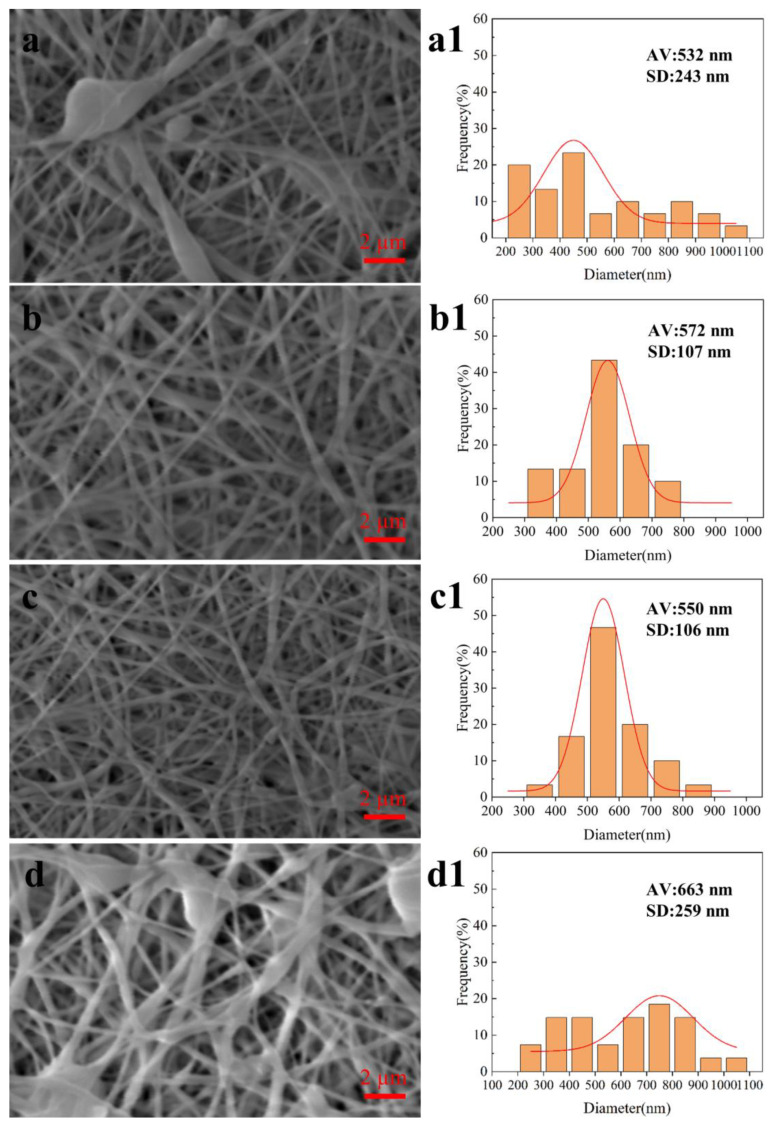
SEM images and fiber diameter distribution of coaxial electrospun membranes prepared with receiving distance. (**a**,**a1**) 12 cm; (**b**,**b1**) 15 cm; (**c**,**c1**) 18 cm; (**d**,**d1**) 21 cm.

**Figure 6 biomimetics-08-00419-f006:**
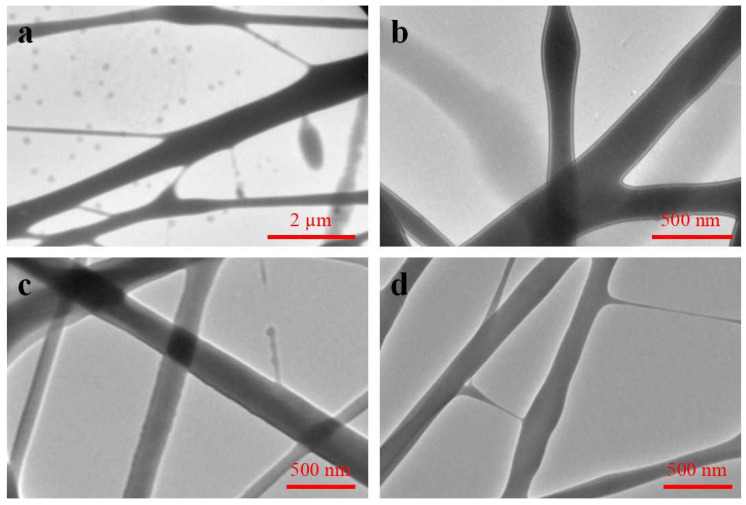
TEM images of coaxial electrospun membranes prepared with different ratios of liquid supply velocity. (**a**) 1:1; (**b**) 1:2; (**c**) 1:3; (**d**) 1:4.

**Figure 7 biomimetics-08-00419-f007:**
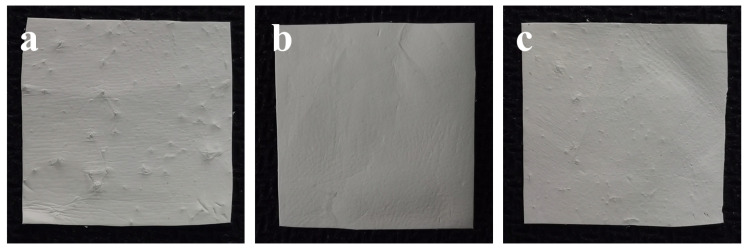
The macroscopic characterization diagram of coaxial electrospun membranes prepared under different key process parameters. (**a**) A_2_B_1_C_3_D_3_; (**b**) A_2_B_2_C_2_D_2_; (**c**) A_2_B_3_C_1_D_1_.

**Figure 8 biomimetics-08-00419-f008:**
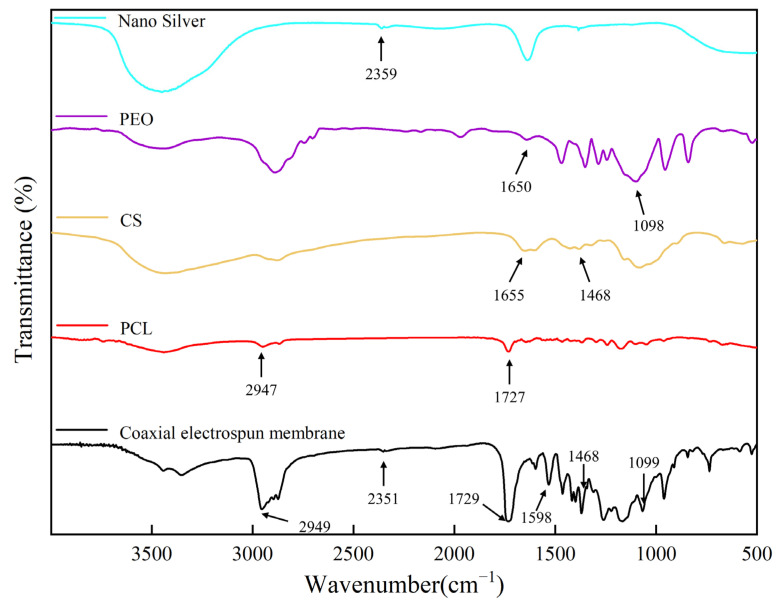
FTIR spectra of Nano-silver, PEO, CS, PCL and coaxial electrospun membrane.

**Figure 9 biomimetics-08-00419-f009:**
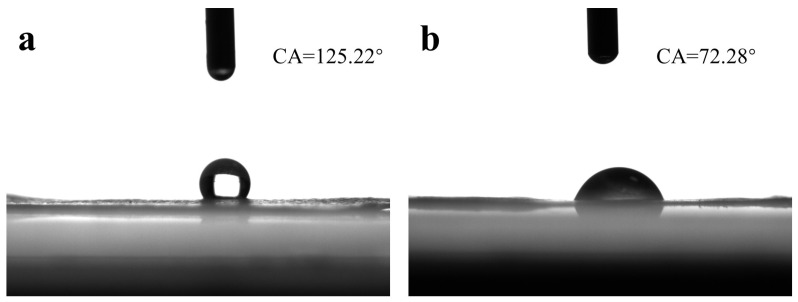
Water contact Angle test results of the membranes. (**a**) PCL membrane; (**b**) coaxial electrospun membrane.

**Figure 10 biomimetics-08-00419-f010:**
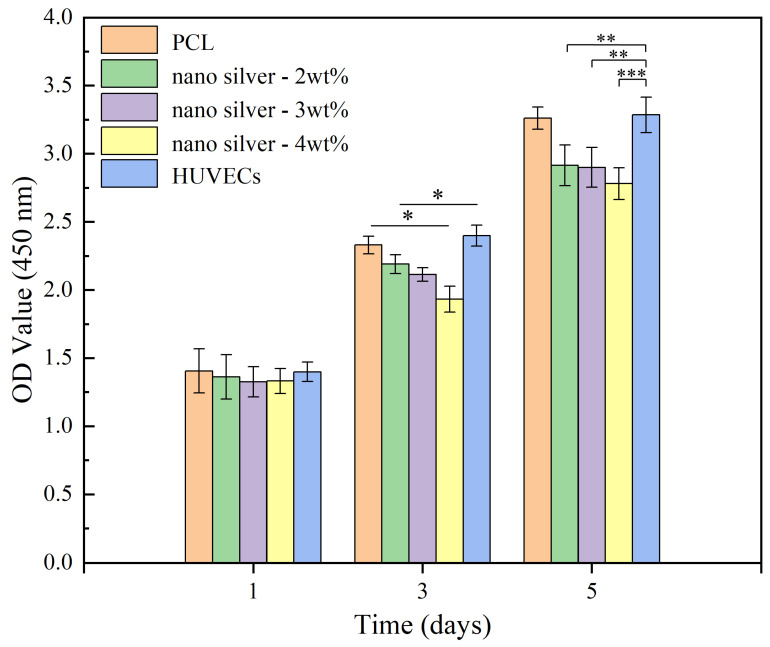
Statistical diagram of cytotoxicity test results. (* *p* < 0.05, *n* = 3; ** *p* < 0.01, *n* = 3; *** *p* < 0.001, *n* = 3).

**Figure 11 biomimetics-08-00419-f011:**
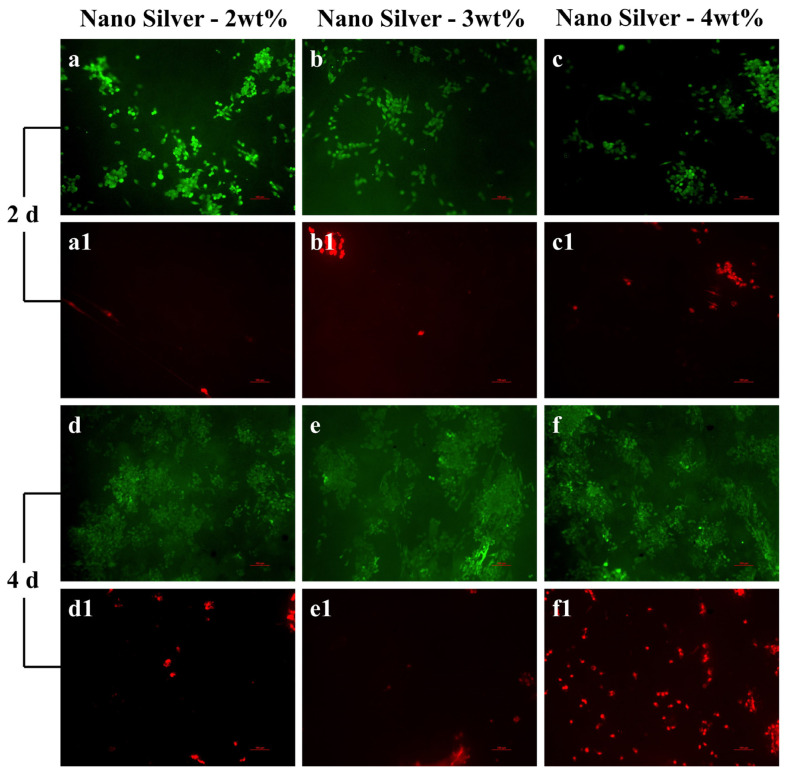
Cell life/death patterns of coaxial electrospun membrane containing different concentrations of nano-silver. (**a**,**a1**,**d**,**d1**) 2 wt%; (**b**,**b1**,**e**,**e1**) 3 wt%; (**c**,**c1**,**f**,**f1**) 4 wt%.

**Figure 12 biomimetics-08-00419-f012:**
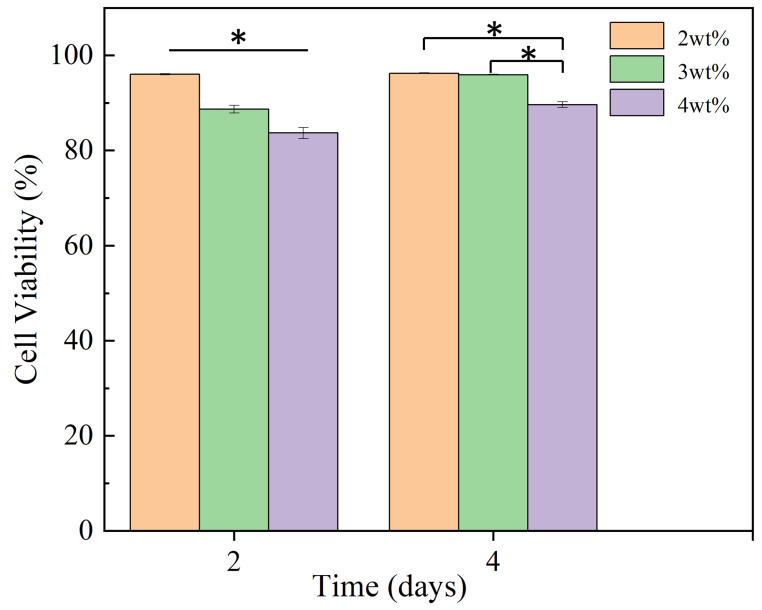
Cell viability of coaxial electrospun membrane containing different concentrations (2 wt%, 3 wt% and 4 wt%) of nano-silver (* *p* < 0.05, *n* = 3).

**Figure 13 biomimetics-08-00419-f013:**
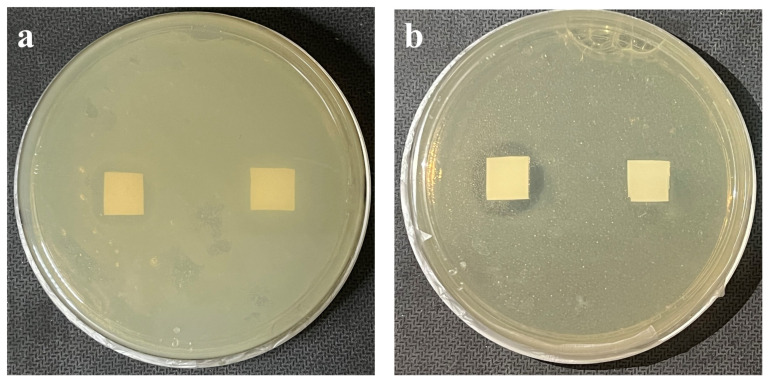
Observation of inhibition area of coaxial electrospun membranes on *E. coli* and *S. aureus*. (**a**) The inhibition area in *E. coli*; (**b**) The inhibition area in *S. aureus*.

**Table 1 biomimetics-08-00419-t001:** Orthogonal test results of key process parameters of coaxial electrospinning.

Experimental Group	Factor				Tensile Strength (MPa)
	A (*w*/*v*)	B (kV)	C (cm)	D (/)	
1	2% (1)	12 (1)	14 (1)	1:2 (1)	1.720 ± 0.106
2	2% (1)	14 (2)	18 (3)	1:3 (2)	2.108 ± 0.132
3	2% (1)	16 (3)	16 (2)	1:4 (3)	2.066 ± 0.143
4	3% (2)	12 (1)	18 (3)	1:4 (3)	2.672 ± 0.127
5	3% (2)	14 (2)	16 (2)	1:3 (2)	2.945 ± 0.092
6	3% (2)	16 (3)	14 (1)	1:2 (1)	2.841 ± 0.071
7	4% (3)	12 (1)	16 (2)	1:3 (2)	2.066 ± 0.146
8	4% (3)	14 (2)	14 (1)	1:4 (3)	2.359 ± 0.092
9	4% (3)	16 (3)	18 (3)	1:2 (1)	1.603 ± 0.084
K_1_	5.894	6.458	6.92	6.164	
K_2_	8.458	7.412	7.077	7.119	
K_3_	6.028	6.51	6.383	7.097	
k_1_	1.965	2.153	2.307	2.055	
k_2_	2.819	2.471	2.359	2.373	
k_3_	2.009	2.17	2.128	2.366	
R	0.854	0.318	0.231	0.318	
Primary and secondary order	A > B = D > C				
Optimal combination	A_2_B_2_C_2_D_2_				

## Data Availability

The data that support the findings of this study are available on request from the corresponding author. The data are not publicly available due to privacy.

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
