# Peer review of "Preparation and Characterization of Nano-Silver-Loaded Antibacterial Membrane via Coaxial Electrospinning"

_biomimetics, 2023, doi:10.3390/biomimetics8050419_

Round 1

Reviewer 1 Report

The comments are attached.

Reviewer 2 Report

The article describe the production of antibacterial membranes through  coaxial electrospinning. The subject is interesting and the analysis of the different parameters to be monitored in the process is adequately described . There are some flaws; 

Page 2 line 56; the sentence should be rewritten

page 2 line 65; broad

page 2 line 66 avoid repetition of antibacterial drug

page 2 line 69; rewrite the sentence

Page 2 line 73; not seriously but Actually

Page 2 line 74 remove open

page 2 line 95 what is dimethyl for artificial purposes?

page 3 line 102: why respectively?

page 3 line 109 and 110 remove the

page 5 line 164, what is CCK-8 ? it is clears they are some kind of cells but it should be said

page 5 168-171: the procedure is not clear

page 5 line 198; cannot imagine the meaning of reporting a measure of 537.67 nanometers. How did the authors measure it which such a high precision? numbers should be reconsidered. This is the same for any measure reported.

Page 15 line 337: E.Coli is not a negative bacteria, it is a Gram-negative bacteria, the opposite for the S. Aureus

Page 15, lines 346-348: it is not clear the meaning of the sentence

Page 16; figure 12. Sincerely the inihibition area in the escerichia Coli experiment is not so evident. the second piece of tissues is the control experiment? in case it should be said

The english used throughout the work is grammarly correct but the sentences are often difficult to understand. This made the reading slow and requires an extra effort to fully understand the content. I would suggest a reconsideration of the writing style

Reviewer 3 Report

This paper examines how different process parameters impact the creation of a membrane using coaxial electrospinning. The membrane, made with CS, PEO, and nanosilver in the core layer and PCL in the shell layer, showed excellent hydrophilicity, biocompatibility, and antibacterial properties against Escherichia coli and Staphylococcus aureus. This membrane has potential for clinical use in vascular stent membranes and bionic blood vessels.

Please find below my comments and suggestions:

Abstract: The length and the content is appropriate. However the use of acronyms is usually not recommended here. Please define the term CS, PEO, PCL, SEM and TEM. Although most of them are quite well-known the first time that they are used it is recommended to give the full word. 

The scope of the paper, the introduction and the novelty are clear. 

Experimental section.

Section 2.1. Dimethylene chloride (DMC) is proposed as solvent. 

Given that dichloromethane is a halogenated solvent, which is not environmentally friendly, it is essential to consider alternative solvents for biomedical applications to ensure sustainability and reduce environmental impact. Are there any other solvents that can be used in the coaxial electrospinning process to achieve the same or better results for the preparation of the membrane with antibacterial properties?

Section 2.2. Regarding the electrospinning conditions, please specify the diameter of the drum collector. 

Section 2.3. Referred to ortogonal analysis, please specify a definition of all the terms used for the analysis so that the any reader can easily follow the design. 

Section 2.4. Mechanical test. If posible indicate how many specimens were tested per sample. 

Results and discussion. In Figure 3 and Figure 4, the labels (a1, b1, c1 and d1) interphere with the graphs. In the fiber diameter distributions, the function used for the fit do not start in 0.  

In the same figures, the inset with the average diameter and standard deviation, it is recommended to round it off and include the units. 

Experimental conditions used for the visualization of the samples by scanning electron microscopy (SEM) are not mentioned neither in the experimental section nor in any of the figures in the results and discussion section. This information is crucial for the reproducibility and credibility of the research findings. 

In the abstract it is mentioned the use of TEM (transmission electron microscopy), but no images are finally included in this document. 

In Table 1, the results of Tensile Strength are used to discuss the performance of the different samples. First, I would like to suggest the authors to include standard deviation of the data provided in the table or any other parameter to characterize the dispersion of the results. 

In addition to analyzing the tensile strength, it would be beneficial to include other mechanical parameters like elastic modulus or deformation to fracture.

Please check formatting in figures. For example in Figure 9, time (days) and Figure 11 time (d). 

Regarding water contact angle (WCA) measurements, it would be interesting also to provide the value of WCA of pure PEO or CS/PEO system for comparsion purposes with those of the coaxial fibers. 

The conclusions of the research work are clear and for that reason, I consider that the paper can be accepted for publication after carefully revision of the manuscript. 
